# CONTEXTUAL SYMBOLIC POLICY FOR META-REINFORCEMENT LEARNING

## ABSTRACT

Context-based Meta-Reinforcement Learning (Meta-RL), which conditions the RL agent on the context variables, is a powerful method for learning a generalizable agent. Current context-based Meta-RL methods often construct their contextual policy with a neural network (NN) and directly take the context variables as a part of the input. However, the NN-based policy contains tremendous parameters which possibly result in overfitting, the difficulty of deployment and poor interpretability. To improve the generation ability, efficiency and interpretability, we propose a novel Contextual Symbolic Policy (CSP) framework, which generates contextual policy with a symbolic form based on the context variables for unseen tasks in meta-RL. Our key insight is that the symbolic expression is capable of capturing complex relationships by composing various operators and has a compact form that helps strip out irrelevant information. Thus, the CSP learns to produce symbolic policy for meta-RL tasks and extract the essential common knowledge to achieve higher generalization ability. Besides, the symbolic policies with a compact form are efficient to be deployed and easier to understand. In the implementation, we construct CSP as a gradient-based framework to learn the symbolic policy from scratch in an end-to-end and differentiable way. The symbolic policy is represented by a symbolic network composed of various symbolic operators. We also employ a path selector to decide the proper symbolic form of the policy and a parameter generator to produce the coefficients of the symbolic policy. Empirically, we evaluate the proposed CSP method on several Meta-RL tasks and demonstrate that the contextual symbolic policy achieves higher performance and efficiency and shows the potential to be interpretable.

## 1 INTRODUCTION

Meta-Reinforcement Learning (Meta-RL) is a promising strategy to improve the generalization ability on unseen tasks of reinforcement learning. Meta-RL methods learn the shared internal structure of tasks from the experiences collected across a distribution of training tasks and then quickly adapt to a new task with a small amount of experiences. On this basis, context-based Meta-RL methods (Duan et al., 2016; Rakelly et al., 2019; Fakoor et al., 2019; Huang et al., 2021) are proposed with the motivation that only a part of the model parameters need to be updated in a new environment. They force their model to be conditional on a set of task-specific parameters named context variables which are formed by aggregating experiences. Context-based Meta-RL methods are attractive because of their empirically higher performance and higher efficiency compared with previous methods which update the whole model.

However, how to incorporate the context variables into the policy is still an open problem. Most of the current methods construct their contextual policy with a neural network (NN) and directly take the context variables as a part of the input. This kind of NN-based policy usually involves thousands of parameters, which may bring training difficulties, possibly result in overfitting and hurt the generalization performance. In addition, deploying the complex NN-based policy is inefficient and even impossible on limited computational resources. What is worse, we have to treat the NN-based policy as a black box that is hard to comprehend and interpret, e.g., we cannot understand what the difference between the policies of different tasks is.

To address the above issues, in this work, we propose a novel Contextual Symbolic Policy (CSP) framework to learn a contextual policy with a compact symbolic form for unseen tasks in meta-RL. We are inspired by the symbolic expression, which has a compact form but is capable of capturing complex relationships by composing variables, constants and various mathematical operators. In general, compact and effective representations can strip out irrelevant information and find the essential relationship of variables, which can benefit the generalization. Therefore, for meta-RL tasks with similar internal structures, CSP produces symbolic policies to model the relationship of the proper action and state and extract essential common knowledge across the tasks. With the common knowledge of a series of tasks, CSP is able to achieve higher generalization ability and quickly adapt to unseen tasks. Moreover, the compact symbolic policies learned by CSP are efficient to be deployed and easier to understand. In conclusion, contextual policies produced by CSP achieve higher generalization performance, efficiency, and show the potential to be interpretable when constrained in a compact symbolic form.

However, finding the proper forms and constant values of the symbolic policies for a distribution of tasks is challenging. In this paper, we propose an efficient gradient-based learning method for the CSP framework to learn the contextual symbolic policy from scratch in an end-to-end differentiable way. To express the policy in a symbolic form, the proposed CSP consists of a symbolic network, a path selector and a parameter generator. The symbolic network can be considered as a full set of the candidate symbolic policies. In the symbolic network, the activation functions are composed of various symbolic operators and the parameters can be regarded as the coefficients in the symbolic expression. For a new task, the path selector chooses the proper compact symbolic form from the symbolic network by adaptively masking out most irrelevant connections. Meanwhile, the parameters of the chosen symbolic form are generated by the parameter generator. We design all these modules to be differentiable. Thus, we can update the whole framework with gradient efficiently.

We evaluate the proposed CSP on several Meta-RL tasks. The results show that our CSP achieves higher generalization performance than previous methods while reducing the floating point operations (FLOPs) by 2-45000×. Besides, the produced symbolic policies show the potential to be interpretable.

## 2 RELATED WORKS

### 2.1 META-REINFORCEMENT LEARNING

Meta-RL extends the notion of meta-learning (Schmidhuber, 1987; Bengio et al., 1991; Thrun & Pratt, 1998) to the context of reinforcement learning. Some works (Li et al., 2017; Young et al., 2018; Kirsch et al., 2019; Zheng et al., 2018; Sung et al., 2017; Houthooft et al., 2018) aim to meta-learn the update rule for reinforcement learning. We here consider another research line of works that meta-train a policy that can be adapted efficiently to a new task. Several works (Finn et al., 2017; Rothfuss et al., 2018; Stadie et al., 2018; Gupta et al., 2018; Liu et al., 2019) learn an initialization and adapt the parameters with policy gradient methods. However, these methods are inefficient because of the on-policy learning process and the gradient-based updating during adaptation. Recently, context-based Meta-RL achieve higher efficiency and performance. PEARL (Rakelly et al., 2019) proposes an off-policy Meta-RL method that infers probabilistic context variables with experiences from new environments. ADARL (Huang et al., 2021) characterizes a compact representation about changes of environments with a structural environment model, which enables efficient adaptation. Hyper (Sarafian et al., 2021) proposes a hypernetwork where the primary network determines the weights of a conditional network and achieves higher performance. Fu et al. (2020) introduce a contrastive learning method to train a compact context encoder. They also train an exploration policy to maximize the information gain. Most of the existing context-based Meta-RL methods(Fu et al., 2020; Zhang et al., 2021; Zintgraf et al., 2020) attempt to achieve higher performance by improving the context encoder or the exploration strategy. However, in this paper, we aim to improve the efficiency, interpretability and performance by replacing the pure neural network policy with a contextual symbolic policy.

## 2.2 SYMBOLIC REGRESSION

Symbolic regression aims to find symbolic expressions to best fit the dataset from an unknown fixed function. Although this problem is likely NP-hard in principle, several works attempt to solve it with heuristic algorithms. Koza (1993) introduce genetic programming (GP) to evolve the symbolic expressions and a series of following works (Schmidt & Lipson, 2010; Cava et al., 2019; Virgolin et al., 2019; de França & Aldeia, 2021) expand the basic genetic programming method to improve the performance. Recently, some methods involve deep learning for symbolic regression. AI Feynman (Udrescu & Tegmark, 2019) utilizes neural networks to discover hidden simplicity in the dataset and break harder problems into simpler ones. DSR (Petersen et al., 2021) train a recurrent neural network (RNN) with reinforcement learning to produce the symbolic expression. Differentiable symbolic regression methods (Martius & Lampert, 2016; Sahoo et al., 2018) use a neural network whose activation functions are symbolic operators as a symbolic expression and decrease the length of symbolic expressions with $L_1$ regularization. However, the plain structure and $L_1$ regularization may fail with complex problems. We also employ a symbolic network for the differentiability. The difference is that we propose a densely connected symbolic network and probabilistic path selector, which enable symbolic meta-policy learning. Besides, these methods are designed for regression of a fixed function while we aim to learn a meta-policy which is conditional on the context variables.

Recently, some works employ symbolic regression methods to obtain symbolic policies for efficiency and interpretability. Kubalík et al. (2017) and Hein et al. (2018) aim to approximate a symbolic policy with genetic programming but require a given dynamics equations or a learned world model. DSP (Larma et al., 2021) employ a recurrent neural network to generate the symbolic policy. They use the average returns of the symbolic policies as the reward signal and train the neural network with risk-seeking policy gradients. However, for environments with multidimensional action spaces, they need a pre-trained neural network policy as the anchor model. Besides, in this framework, a single reward for reinforcement learning involves many environmental interactions, which is inefficient and makes it hard to combine the symbolic policy with Meta-RL. Recently, some works (Bastani et al., 2018; Verma et al., 2018) attempt to distill an interpretable policy from a pre-trained neural network policy but have a problem of objective mismatch (Larma et al., 2021). Different from the methods talked above, we propose an efficient gradient-based framework to obtain the symbolic policy without any pre-trained model. As far as we know, we are the first to learn a symbolic policy from scratch and use the symbolic policy for Meta-RL. There exist other works introduce the symbolic to reinforcement learning. Garnelo et al. (2016); d'Avila Garcez et al. (2018) learn the symbolic representation for better interpretability. Lyu et al. (2019) introduces symbolic planning for efficiency and interpretability. In this paper, we focus on learning the symbolic policy for Meta-RL.

## 3 PRELIMINARIES

In the field of meta-reinforcement learning (Meta-RL), we consider a distribution of tasks $p(\kappa)$ with each task $\kappa \sim p(\kappa)$ modeled as a Markov Decision Process(MDP). In common Meta-RL settings, tasks share similar structures but differ in the transition and/or reward function. Thus, we can describe a task $\kappa$ with the 6-tuple $(\mathcal{S}, \mathcal{A}, \mathcal{P}_\kappa, \rho_0, r_\kappa, \gamma)$. In this setting, $\mathcal{S} \subseteq \mathbb{R}^n$ is a set of n-dimensional states, $\mathcal{A} \subseteq \mathbb{R}^m$ is a set of m-dimensional actions, $\mathcal{P}_\kappa : \mathcal{S} \times \mathcal{A} \times \mathcal{S} \to [0, 1]$ is the state transition probability distribution, $\rho_0 : \mathcal{S} \to [0, 1]$ is the distribution over initial states, $r_\kappa : \mathcal{S} \times \mathcal{A} \to \mathbb{R}$ is the reward function, and $\gamma \in (0, 1)$ is the per timestep discount factor. Following the setting of prior works (Rakelly et al., 2019; Fakoor et al., 2019), we assume there are $M$ meta-training tasks $\{\kappa_m\}_{m=1,\cdots,M}$ sampled from the training tasks distribution $p_{train}(\kappa)$. For meta-testing, the tasks are sampled from the test tasks distribution $p_{test}(\kappa)$. The two distributions are usually same in most settings but can be different in out-of-distribution(OOD) settings. We denote context $c_T = \{(s_1, a_1, s'_1, r_1), \cdots, (s_T, a_t, s'_T, r_T)\}$ as the collected experiences. For context-based Meta-RL, agent encodes the context into a latent context variable $z$ with a context encoder $q(z|c_T)$ and the policy $\pi$ is conditioned on the current state and the context variable $z$. During adaptation, agent first collect experiences for a few episodes and then update the context variables and maximize the return with the contextual policy. The Meta-RL objective can be formulated as $\max_\pi \mathbb{E}_{\kappa \sim p(\kappa)}[\mathbb{E}_{c_T \sim \pi}[R(\kappa, \pi, q(z|c_T))]]$, where $R(\kappa, \pi, q(z|c_T))$ denotes the expected episode return.

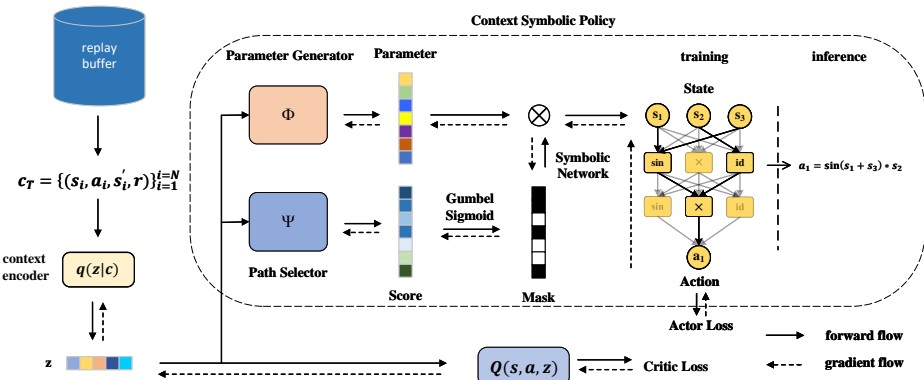

Figure 1: Illustrations of our framework. The Contextual Symbolic Policy (CSP) produces different symbolic policies based on the context variables across tasks. CSP construct the symbolic policy with the symbolic network, selects the proper symbolic form with the Path Selector and generate the parameters of symbolic policy with the Parameter generator. The whole framework is differentiable and can be learned end-to-end.

## 4 GRADIENT-BASED CONTEXTUAL SYMBOLIC POLICY

This section introduces the structure of our contextual symbolic policy, an end-to-end differentiable system that can be directly updated with gradient. As Figure 1 shows, the contextual symbolic policy consists of three main components: 1) the Symbolic Network, which expresses the policy in a symbolic form, 2) the Parameter Generator, which outputs the parameters for the symbolic network according to the context variables, and 3) the Path Selector, which select paths from the symbolic network to form compact symbolic expressions.

### 4.1 DENSELY CONNECTED SYMBOLIC NETWORK

To construct a symbolic policy in an end-to-end differentiable form, we propose the densely connected symbolic network. Inspired by previous differentiable symbolic regression methods (Martius & Lampert, 2016; Sahoo et al., 2018), we employ a neural network with specifically designed units, which is named symbolic network. We now introduce the basic symbolic network named plain structure which is illustrated in Figure 2. The symbolic network is a feed-forward network with $L$ layers. Different from traditional neural networks, the activation functions of the symbolic network is replaced by symbolic operators, e.g. trigonometric functions and exponential functions. For the $l_{th}$ layer of the symbolic network, we denote the input as $x_{l-1}$ and the parameters as $w_l, b_l$. These parameters serve as the constant in a symbolic expression. We assume that the $l_{th}$ contains $m$ unary functions $\{g_1^1, \cdots, g_m^1\}$ and $n$ binary functions $\{g_1^2, \cdots, g_n^2\}$. Firstly, the input of the $l_{th}$ layer will be linearly transformed by a fully-connected layer:

$$y = F_l(x) = w_l x + b_l. \tag{1}$$

The fully-connected layer realizes the addition and subtraction in symbolic expressions and produces $m + 2n$ outputs. Then the outputs will go through the symbolic operators and be concatenated to form the layer output:

$$G_l(y) = [g_1^1(y_1), \cdots, g_m^1(y_m), g_1^2(y_{m+1}, y_{m+2}), \cdots, g_n^2(y_{m+2n-1}, y_{m+2n})] \tag{2}$$

Then the $l_{th}$ layer of the symbolic network can be formulated as $S_l : x_l = G_l(F_l(x_{l-1}))$. Following the last layer, there will be a fully-connected layer to produce a single output. For multiple action dimensions, we construct a symbolic network for each dimension of action.

**Symbolic operator.** The symbolic operators are selected from a library $\mathcal{L}$, e.g. $\{sin, cos, exp, log, \times, \div\}$ for continuous control tasks. For the plain structure, we include an identical operator which retains the output of the previous layer to the next layer in the library. We also provide an optional conditional operator $c(a, b, c) = sigmoid(a) * b + (1 - sigmoid(a)) * c$ to approximate **if** A **then** B **else** C. We find it is useful in some tasks. Note that we aim to find the symbolic policy with gradient. Thus, it is critical to ensure the numerical stability of the

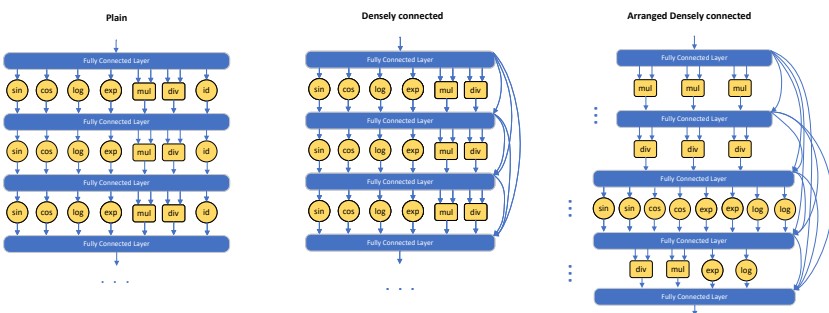

Figure 2: Example network structures for the symbolic network. **Left**: the plain structure. **Middle**: a symbolic work with dense connections. **Right**: a symbolic network with dense connections and arranged operators.

system. However, this is not natural in a symbolic network. For example, the division operator and the logarithmic operator will create a pole when the input goes to zero and the exponential function may produce a large output. Thus, we regularize the operators and employ a penalty term to keep the input from "forbidden" area. For example, the logarithmic operator $y = log(x)$ returns $log(x)$ for $x > bound_{log}$ and $log(bound_{log})$ otherwise and the penalty term is defined as $\mathcal{L}_{log} = max(bound_{log} - x, 0)$. The division operator $c = a/b$ returns $a/b$ for $b > bound_{div}$ and $0$ otherwise. The penalty term is defined as $\mathcal{L}_{div} = max(bound_{div} - b, 0)$. The details of all regularized operators can be found in the Appendix.

**Dense connectivity.** We introduce dense connections (Huang et al., 2017) in the symbolic network, where inputs of each layer are connected to all subsequent layers. Consequently, the $l_{th}$ layer of the symbolic network will receive the environment state $s$ and the output of all preceding layers $x_1, \cdots, x_{l-1}$: $x_l = G_l(F_l([s, x_1, \cdots, x_{l-1}]))$. On the one hand, the dense connections improve the information flow between layers and benefit the training procedure. On the other hand, the dense skip connections across layers enable us to control the complexity of the symbolic expressions with the parameters more flexibly. In practice, each layer in the symbolic network may contain different operators. With the dense connectivity, we can flexibly arrange the position of operators. For example, if we only arrange the $sin$ operator in the last layer but the oracle expression contains terms like $sin(s_0)$, the input of the $sin$ operator can still be from the original state because of the dense connections. We give an example of arranged operators in Figure 2 which we use for all tasks in the experiments. In this symbolic network, we assume that multiplication and division operations are more likely to occur at shallow layers, while more complex operations such as sines and cosines are more likely to occur at deep layers. In addition, we avoid the form $sin(\cdots exp(\cdots)\cdots)$ which rarely occurs in physics formulas with the arrangement.

## 4.2 INCORPORATING THE CONTEXT VARIABLES

To produce different symbolic policies with the symbolic network for different tasks $\kappa$ sampled from the task distribution $p(\kappa)$, we need to incorporate the context variables $z \sim q(z|c_T)$ to the symbolic network. To condition the parameters of the symbolic expression on the context variable, we propose a parameter generator: $\boldsymbol{w_g} = \Phi(z)$ which is a neural network to produce the parameters of symbolic networks for all action dimensions based on the context variables.

However, the symbolic network serves as a full set of the search space of symbolic expressions. To select the proper paths from the symbolic network to produce a compact symbolic policy, we reduce the number of paths involved in the final symbolic policy then proper paths remain and redundant paths are removed. This can be naturally realized by minimizing the $L_0$ norm of the symbolic network parameters. As the $L_0$ norm is not differentiable, some methods (Martius & Lampert, 2016; Sahoo et al., 2018) employ $L_1$ norm instead of $L_0$ norm. However, $L_1$ will penalize the magnitude of the parameters. In our framework, the parameters of the symbolic network is the output of a neural network rather than independent variables. The penalty of magnitude may severely affect the training of the parameter generator. Inspired by the probability-based sparsification method (Srinivas et al., 2017; Louizos et al., 2017; Zhou et al., 2021), we propose a probabilistic path selector which selects path from the network by multiplying a binary mask on the parameters of the symbolic network. The path selector first produce scores with the context variables: $\boldsymbol{s} = \Psi(z)$,

where $s_i \in (0, 1)$. The score $s_i$ serves as the probability of the Bernoulli distribution and the binary mask $m_i$ is sampled from the distribution: $m_i \sim Bern(s_i)$. Then the final parameters of the symbolic network are $\boldsymbol{w} = \boldsymbol{w_g} \bigotimes \boldsymbol{m}$, where $\bigotimes$ is the element-wise multiply operation. Consequently, to get a compact symbolic expression, we only need to minimize the expectation of the $L_0$ norm of the binary mask $\mathbb{E}_{\boldsymbol{m} \sim p(\boldsymbol{m}|\boldsymbol{s})} \|\boldsymbol{m}\|_0 = \sum s_i$, without penalizing the magnitude of the parameters.

During the process of collecting data or testing, we can directly sample the binary mask from the Bernoulli distribution. However, the sampling process does not have a well-defined gradient. Thus, for the training process we build up our sampling function with the gumbel-softmax trick (Jang et al., 2016). As the mask $\boldsymbol{m}$ is binary categorical variables, we replace the *softmax* with *sigmoid* and named the sampling function as *gumbel sigmoid*. The *gumbel sigmoid* function can be formulated as:

$$\boldsymbol{m_{gs}} = sigmoid(\frac{log(\frac{\boldsymbol{s}}{1-\boldsymbol{s}}) + \boldsymbol{g_1} - \boldsymbol{g_0}}{\tau}), \tag{3}$$

where $\boldsymbol{g_1}$ and $\boldsymbol{g_0}$ are i.i.d samples drawn from $Gumble(0, 1)$. $\tau$ is the temperature annealing parameter. Note that $\boldsymbol{m_{gs}}$ is still not a binary mask. To obtain a binary mask but maintain the gradient, we employ the Straight-Through (ST) trick: $\boldsymbol{m} = \mathbb{1}_{\geq 0.5}(\boldsymbol{m_{gs}}) + \boldsymbol{m_{gs}} - \overline{\boldsymbol{m_{gs}}}$, where $\mathbb{1}_{\geq 0.5}(x) \in \{0, 1\}^n$ is the indicator function and the overline means stopping the gradient. With the path selector, the framework is able to produce very short symbolic policies for relatively simple tasks while produce complex but compact symbolic policies to handle hard tasks like Walker2d and Hopper in Mujoco Simulator(Todorov et al., 2012).

## 5 META-LEARNING THE SYMBOLIC POLICY

In this section, we introduce the meta-learning process of our symbolic policy. Following PEARL (Rakelly et al., 2019), we build up our off-policy learning framework on top of the soft actor-critic algorithm (SAC) (Haarnoja et al., 2018). The main differences are the additional loss for the symbolic policy and the schedule for collecting data and simplifying symbolic expressions.

### 5.1 LOSS FUNCTION

We now illustrate our additional loss function for the symbolic policy. As described in Section 4.1, to ensure the numerical stability, we regularize the symbolic operators and employ a penalty term for regularized operators. During training, we involve a penalty loss function $\mathcal{L}_{penalty}$ which is the sum of the penalty terms of regularized operators in symbolic networks:

$$\mathcal{L}_{penalty}(\theta_\Phi, \theta_\Psi) = \sum_{i=1}^{i=M} \sum_{j=1}^{j=L} \sum_{k=1}^{k=N_j} \mathcal{L}_{g_{i,j,k}}(x_{i,j,k}), \tag{4}$$

where $\theta_\Phi, \theta_\Psi$ is the parameters of the parameter generator and the path selector, $M$ is the dimension of action, $L$ is the number of layers in a symbolic network, $N_j$ is the number of regularized operators in layer $j$, $x_{i,j,k}$ is the input of operator $g_{i,j,k}$ and $\mathcal{L}_{g_{i,j,k}}$ is the penalty term for this operator. We also involve a loss function $\mathcal{L}_{select}$ to regularize the sum of score $\boldsymbol{s}$ which is the expectation of the $L_0$ norm of the binary mask $\boldsymbol{m}$ as described in Section 4.2. To limit the minimum complexity of symbolic policies, we involve the target $L_0$ norm defined as $l_{target}$. Then the loss function can be defined as:

$$\mathcal{L}_{select}(\theta_\Psi) = max(\sum s_i - l_{target}, 0) \tag{5}$$

### 5.2 TRAINING SCHEDULE

In practice, we train our symbolic policy in an off-policy manner. For meta-training epoch $t$, the agent first collects experiences into the corresponding buffer $\mathcal{B}_{\kappa_i}$ for several iterations. At the beginning of each iteration, we sample context $c_T$ from buffer $\mathcal{B}_{\kappa_i}$ and sample context variables $z \sim q(z|c_T)$ as PEARL does. The difference is that we also sample the symbolic policy with $\Phi(z)$ and $\Psi(z)$ and use the sampled policy for the following steps of the iteration. Then we sample RL batch and context from the buffer and optimize the context encoder $q(z|c_T)$ to recover the state-action value function. For each training step, we sample a new symbolic policy. We employ the

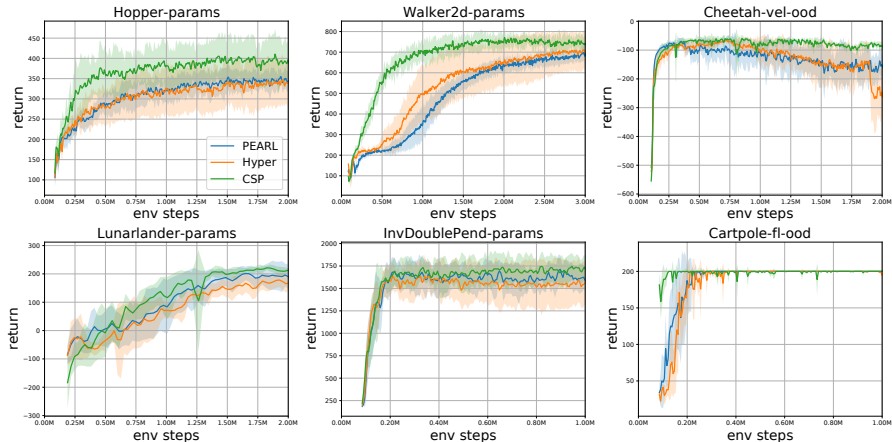

Figure 3: Comparison for different kinds of contextual policies on Meta-RL tasks. We show the mean and standard deviation of returns on test tasks averaged over five runs.

soft actor-critic to optimize the state-action value function. For the parameter generator and the path selector, we employ $\mathcal{L}_{select}$ and $\mathcal{L}_{penalty}$ in addition to the SAC loss. During training, we decrease the temperature parameter $\tau$ of *gumbel sigmoid* linearly and decrease the $l_{target}$ from the length of the origin parameters $\boldsymbol{w_g}$ to a target value with a parabolic function. We illustrate the gradient flow in Figure 1. More details and the pseudo-code can be found in the Appendix.

## 6 EXPERIMENT

In this section, we present the experimental results of our contextual symbolic policy (CSP). We first compare CSP to prior meta policies on several Meta-RL problems to evaluate the performance and the inference efficiency in Section 6.1. Then we analyze the symbolic policy generated for different tasks in Section 6.2. Finally, we carry out ablation experiments for CSP and show the results in 6.3.

### 6.1 COMPARISON OF CONTEXTUAL POLICIES

**Experimental Settings.** We first evaluate CSP on several continuous control environments. We modify the environments of OpenAI Gym (Brockman et al., 2016), including kinds of classic control, Box2D, and MuJoCo (Todorov et al., 2012) to be Meta-RL tasks similar to Rakelly et al. (2019); Huang et al. (2021); Fu et al. (2020). These environments require the agent to adapt across dynamics (random system parameters for Hopper-params, Walker2d-params, Lunarlander-params, InvDoublePend-params, different force magnitude and pole length for Cartpole-fl-ood) or reward functions (target velocity for Cheetah-vel-ood). For the symbolic network, we use the arranged densely connected structure in Figure 2. We run all environments based on the off-policy meta-learning framework proposed by PEARL and use the same evaluation settings. We compare CSP with PEARL which concatenates the observation and context variables as the input of policy and Hyper (Sarafian et al., 2021) which generate the parameters of policy with a ResNet model based on the context variables. Note that the original Hyper also modifies the critic, but we build all the critics with the same network structure for consistency. More details of the environments and hyper-parameters can be found in the Appendix.

**Performance comparisons.** In Figure 3, we report the learning curves of undiscounted returns on the test tasks. We find that in all the environments, CSP achieves better or comparable performance than previous methods. In Hopper-params, Walker2d-params and InvDoublePend-params, CSP outperforms PEARL and Hyper during the whole training process. In Lunarlander-params, CSP achieves better final results. In Cartpole-fl-ood, CSP adapts to the optimal more quickly. In the out-of-distribution task Cheetah-vel-ood, we find the performance of PEARL and Hyper decrease during training because of over-fitting. But our CSP is less affected. In conclusion, expressing the policy in the symbolic form helps improve the generalization performance.

Table 1: FLOPs and inference time of different contextual policies.

| Environment | FLOPs/k | | | Times/ms | | |
|---|---|---|---|---|---|---|
| | CSP | PEARL | Hyper | CSP | PEARL | Hyper |
| Walker2d-params | **3.11** | 189.31 | 5.64 | **20.89** | 27.00 | 22.64 |
| Hopper-params | **0.51** | 186.90 | 4.10 | **4.13** | 26.58 | 17.23 |
| InvDoublePend-params | **0.039** | 186.00 | 3.59 | **0.37** | 25.05 | 12.32 |
| Cartpole-fl-ood | **0.004** | 183.90 | 1.79 | **0.042** | 23.90 | 9.08 |
| Lunarlander-g | **0.015** | 185.4 | 3.08 | **0.l4** | 23.43 | 12.34 |
| Cheetah-vel-ood | **0.53** | 190.21 | 7.18 | **4.90** | 28.44 | 24.16 |

Table 2: Average count of all selected paths and paths selected by at least ninety percent policies.

| Environment | Selected paths | Mostly selected paths |
|---|---|---|
| Walker2d-params | 76.42 | 70.3 |
| Hopper-params | 21.5 | 20.33 |
| InvDoublePend-params | 23.2 | 21.0 |
| Cartpole-fl-ood | 3.06 | 3.0 |
| Lunarlander-g | 5.2 | 5.0 |
| Cheetah-vel-ood | 27.04 | 18.5 |

**Efficiency comparisons.** We also evaluate the deploying efficiency of contextual policies. We first calculate the flops of each kind of policy per inference step. Then we consider an application scenario that the algorithm control five thousand simulated robots with the Intel(R) Xeon(R) Gold 5218R @ 2.10GHz CPU and record the elapsed time per inference step [1]. We report the results in Table 1. Compared to PEARL, CSP reduces the FLOPs by 60-45000x and reduces the inference time by up to 600x. Compared to Hyper, CSP reduces the flops by 2-450x and reduces the inference time by up to 200x. Thus, compared with pure neural network policies, the contextual symbolic policy has a significant advantage in computational efficiency.

## 6.2 ANALYSIS OF SYMBOLIC POLICIES

We then analyze the symbolic policies for different tasks produced by CSP. For each environment, we sample 10 tasks from the environment task distribution and obtain the corresponding symbolic policies with CSP. Then we analyze the selected paths of these policies which determine the forms of the symbolic expressions. Table 2 shows the results. We calculate the average count of selected paths per action dimension among the policies[2]. We find that this number varies across different environments. The symbolic expression can be extremely short for simple environments like Cartpole or relatively long for complex environments like Walker2D. We also calculate the average count of paths which are selected by more than ninety percent of the symbolic policies. In almost all environments, the mostly selected paths account for a high percentage of the selected paths, which indicates that the expressions of symbolic policies for different tasks of the same environment share similar forms.

The proposed CSP can also improve interpretability. We take the Cartpole-fl-ood environment as an example and illustrate the Cartpole system in Figure 4. The form of the symbolic policies produced by CSP is $action = c1 * \theta + c_2 * \dot{\theta} + b$, where $\theta$ is the the angle of the pole and $\dot{\theta}$ is the rotation rate of the pole. $c1$ and $c2$ are the positive coefficients and $b$ is a small constant which can be ignored. The action is the force scale to push the cart. Then the policy can be interpreted as pushing the cart in the direction that the pole is deflected or will be deflected. To analyze the difference between policies for different tasks, we uniformly set the force magnitude and the length of the pole. Then we generate the symbolic policy with CSP and record the coefficients. As Figure 5 shows, $c1$ and $c2$ tend to increase when the force magnitude decrease and the length increase, which is in accord with our intuition. We will give examples of symbolic policy for other environments in the Appendix.

---

[1] Note that we accelerate Hyper by only updating the parameters of policy when we update the context variables. Thus, Hyper only uses the policy model without the ResNet model to infer an action.

[2] We only consider paths that contribute to the final expression.

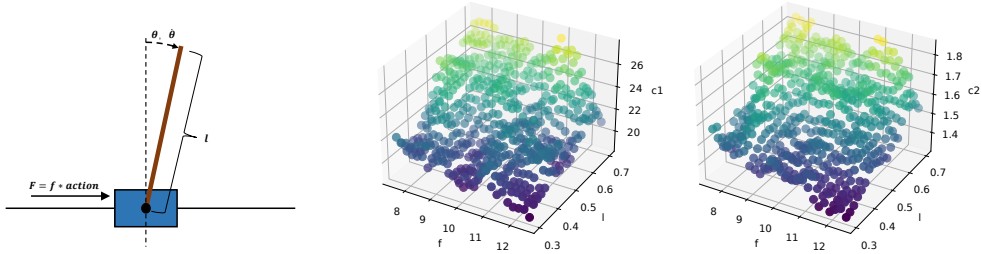

Figure 4: The Cartpole system to be controlled.

Figure 5: The coefficients of symbolic policies for Cartpole environments with different force magnitude and pole length.

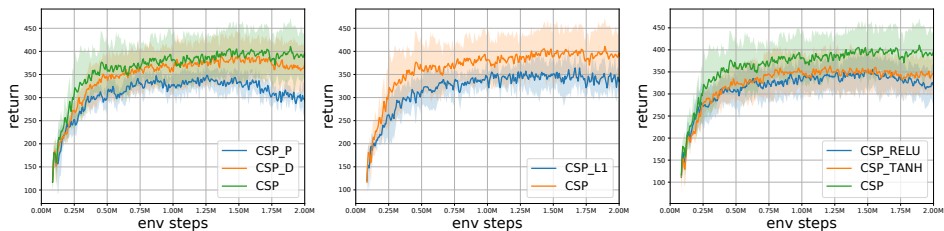

Figure 6: Ablation results of the symbolic network structure (left), the path selector(middle) and the symbolic operator(right). CSP_P means the plain structure and CSP_D means the densely connected structure. CSP_L1 means replacing the path selector with the $L_1$ norm minimization. CSP_TANH means replace all the symbolic operators with *tanh*. CSP_RELU means replace all the symbolic operators with *relu*.

### 6.3 ABLATION

Finally, we carry out experiments by ablating the features of CSP. We first examine our choice of the symbolic structure. We replace the symbolic network with a plain structure and a densely connected structure and compare the test task performance on Hopper-params environments. As Figure 6 shows, the dense connections effectively improve the performance and we can facilitate the search for the proper symbolic form by arranging the operators to further improve the performance. We also replace our path selector with the $L_1$ norm minimization. For the stability of training, we linearly increase the scale of the $L_1$ loss. To get a compact symbolic expression, we set the parameters with the absolute value less than 0.01 as zero near the end of training. We carry out experiments on the Hopper-params and show the learning curves of return in Figure 6. Besides, we calculate the average $L_0$ norm of the mask for our path selector and the count of non-zero parameters for $L_1$ norm minimization which is 30.38 and 34.59, respectively. Compared with the $L_1$ norm minimization, our path selector achieves higher performance when producing slightly more compact symbolic policies. We also replace the symbolic operators with commonly used activation functions *tanh* and *relu*. We use the same framework to select the proper paths and set the same final $L_1$ norm. The results in Figure 6 show that by combine different operators to form the symbolic policy, CSP are able to handle complex relationship between action and state and achieve higher performance compared with single kind of operators.

### 7 CONCLUSION

In this paper, we propose to learn a contextual symbolic policy for Meta-RL. In our gradient-based learning framework, we train the contextual symbolic policy efficiently without any pre-trained model. The contextual symbolic policy achieves higher generalization performance than previous methods. Besides, it is more efficient when deployed and has better interpretability. Our approach may inspire future works of symbolic policy for reinforcement learning or meta-reinforcement learning.

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
