# OpenReview forum: "Contextual Symbolic Policy For Meta-Reinforcement Learning"
_ICLR.cc/2023/Conference — Submitted to ICLR 2023_

### Official Review · Reviewer_EqMi · 2022-10-23

**Confidence:** 3
**Correctness:** 2
**Technical Novelty And Significance:** 2
**Empirical Novelty And Significance:** 3
**Recommendation:** 5

**Clarity, Quality, Novelty And Reproducibility:**

# Detailed Comments (Clarity, Quality, Novetly, Reproducibility)

-   Section 1: "Context-based Meta-RL methods are attractive because of
    their empirically higher performance and higher efficiency compared
    with previous methods which update the whole model." It is not clear
    what "previous methods" are being compared to against here. Do you
    mean RNN based approaches (Duan et al. 2016)?

-   Section 1, "This kind of NN-based policy usually involves thousands
    of parameters, which may bring training difficulties, possibly
    result in overfitting and hurt the generalization performance"

    Thousands of parameters is not an inordinate amount -
    over-parameterization is the de-facto approach. It is also not
    evident that this hurts generalization.

-   Section 4.1: "Different from traditional neural networks, the
    activation functions of the symbolic network is replaced by symbolic
    operators, e.g. trigonometric functions and exponential functions."

    I do not see how these are different from activation functions,
    because an activation function can be any non-linearity. Only that
    sinusoids and exponentials are not commonly used activation
    functions.

-   Section 4.1 (Plain structure vs densely connected): It is not clear
    how a plain structure differs from the densely connected structure,
    given that the plain structure has an identity operator which allows
    skip connections. Also, minor, but densely connected structure is a
    misnomer, as a plain structure seems closer to a traditional dense
    layer in a neural network and your "densely connected" structure
    introduces skip connections.

-   Section 4.1 (Symbolic Networks in general): I do not see much
    difference between symbolic networks and regular feed-forward neural
    networks. There are certain design choices made, such as limiting
    the operations to some library, but this is not much different from
    feed-forward neural networks. In fact, the issues with composing
    arbitrary operations in your library seems more cumbersome for
    little added benefit.

-   Section 4.2: This is essentially a type of hypernetwork, which
    outputs the parameters of the symbolic network, correct? But if Φ is
    a neural network, the the meta-policy is not interpretable and
    suffers from all the issues described previously. While any
    particular policy may be interpretable, that will not provide
    interpratability (or as you claim, generalization) to new tasks -
    which is precisely the type of generalization that matters in meta
    RL.

-   Section 4.2 (path selector vs parameter generator): I do not
    understand the roles of these two components. While it is true that
    you can model a policy using a subset of the symbolic network, it
    does not seem obviously necessary to do so. In fact, this sacrifices
    generalization because you will no longer be sharing parameters at
    the level of the symbolic network. Instead, generalization comes
    from the neural network which generate the parameters and select
    paths - but again this is what you set out to avoid.

-   Section 5.2: "For each training step, we sample a new symbolic
    policy"

    What is the definition of training step here, is it the same as the
    meta-training epoch? It doesn't seem to make sense to resample the
    policy per environment step, and the earlier sentence seems to
    suggest that you "use the sampled policy for the following steps of
    the iteration"

-   Section 6.1: "In Hopper-params, Walker2d-params and
    InvDoublePend-params, CSP outperforms PEARL and Hyper during the
    whole training process"

    While the mean performance is slightly higher, this is not enough to
    claim that CSP outperforms PEARL and Hyper during the whole training
    process. For much of the training process, the confidence intervals
    are overlapping, which suggests that there is not a statistically
    significant difference between them. The results earlier in training
    on WAlker suggest that CSP learns quicker early on.

-   Section 6.1 (Efficiency Comparison): I am somewhat surprised that
    the differences in FLOPs and time can be so significant. Does this
    account for the fact that the CSP is re-sampled

-   Section 6.2 (ablation): From the figure, it does not seem that there
    is a statistically significant difference between CSP with relu,
    tanh or your proposed activation functions,

    Is a CSP with relu or tanh activations functionally equivalent to a
    traditional neural network? If so, it is possible that the neural
    network is not large enough to represent complex nonlinearities that
    CSP provides with its various activation functions.




**Strength And Weaknesses:**

# Strengths

-   An interesting approach to learning interpretable policies. Using
    both a generator and a path selector for learning a symbolic network
    seems new, especially for contextual meta-RL

# Weaknesses

-   The main motivation is unclear. While the paper motivates the need
    for symbolic networks by saying that overparameterized neural
    networks can lead to overfitting and generalization, this is never
    established and the current literature does not support that claim.
    In addition, the way that the symbolic network is learned is, in the
    end, with two neural networks. This seems to further undermine the
    motivation.
-   It is not that obvious how symbolic networks, as described in the
    plain and densely-arranged architecture, differ from neural networks
    aside from differing activation functions. If this is the only
    difference, then this seems like a rather limited distinction from
    regular neural networks. While in continuous control problems, your
    choice of activation functions may lead to a more interpretable
    policy - this may not be the case for some other problem for which
    the needed non-linear activation function is not represented in your
    library.


**Summary Of The Paper:**

To learn symbolic policies that generalize better while maintaining
interpretability, the authors of this paper propose to use neural
networks to generate parameters of a symbolic network which is then
"pruned" through a path selector. Both the generator and the path
selector are conditioned on context variables in the setting of
contextual meta RL. The primary finding is that the learned policies can
be interpretable and generalize better to different contexts, such as
differing mass and forces in a control problem.



**Summary Of The Review:**

# Decision

The goal of this paper is to learn policies for meta-learning that are
more interpretable and more generalizable, This is a valuable goal to
achieve in meta-RL and the research direction, using symbolic networks
that are generated by context, is interesting. While interpretability is
somewhat demonstrated, improvements in generalization are not addressed.
In addition, the symbolic network itself does not seem very different
from a regular neural network that is arranged with special activation
functions. The similarity to regular neural networks, as well as the use
of regular neural networks to generate the symbolic policy, cast doubt
on the motivation and the broad effectiveness of symbolic networks.
Overall, the research direction is not well-motivated and I find that
the paper further undermines the motivation. As a result, I am rating
the paper at a weak reject.

---

### Official Review · Reviewer_KRNC · 2022-10-25

**Confidence:** 3
**Correctness:** 2
**Technical Novelty And Significance:** 2
**Empirical Novelty And Significance:** 2
**Recommendation:** 3

**Clarity, Quality, Novelty And Reproducibility:**

I generally find that the clarity could be significantly improved. In its existing form, many of the high-level ideas and motivation are completely absent or are hidden amongst the details, which makes it difficult to understand this work. For example:
- Why should we expect that "symbolic" operators based on sin, cos., etc. provide greater utility or interpretability than standard NN building blocks, especially when there are already fully connected layers interleaved in between?
- The full algorithm is not precisely defined in the main text, though I note that there is pseudocode in the Appendix, which helps
- Certain details are jumped into without appropriate motivation or justification / explanation, such as the discussion of minimizing the $L_0$ norm. This requires a lot of work on the readers' part to infer the high-level ideas.
- Terms, such as $\mathcal{L}_{g_{i, j, k}}$ are never defined

The proposed ideas do not appear to be particularly novel: ultimately, the proposed approach is a new parametrization of meta-RL policies based on defining paths through "symbolic" operator graphs. Though, I note that I do not find novelty to be one of the most major concerns for this work.

**Strength And Weaknesses:**

This work has two main weaknesses. First, the clarity of the writing and presentation could be significantly improved, and is currently challenging to understand. I elaborate upon that in the next box. Second, this work does not meet the main claims that it sets out of increasing generalization and interpretability. More specifically, the experimental evaluation of these items is not sufficiently thorough:
- It is true that in the reported experiments, the baselines' performance degrades with further training on the "OOD" tasks, while the proposed method achieves high performance throughout training. However, early in training the baselines do still achieve near-optimal results on the OOD test set, and its further performance drop can easily be remedied with early-stopping, which makes this sort of generalization gap less significant. More significant to the community would be considering domains where existing methods cannot already easily generalize out of distribution.
- The claims about interpretability of the system are unclear. It is somewhat interesting that the symbolic policy produces an equation that relates to physics on the cartpole domain, as reported in the experiments. However, can this only happen when there are no dense layers interleaved with the policy? If the "symbolic" operations occur in between dense layers, how can a tight analytic equation be extracted? Additionally, it is unclear to me that compositions of operations such as sin, cos, division, etc., are any more interpretable than large matrix multiplications, especially when such compositions are long. Finally, in image-based domains, such "symbolic" operations seem unlikely to be helpful anyway.
- The experimental evaluation setup could generally be improved to be more informative. Specifically, existing works already evaluate on common MuJoCo benchmarks, and it seems useful to evaluate on those rather than creating new ones. It's unclear to me if the Hopper and Walker evaluations indeed are the ones that have been used in other works, or if they are variants created newly in this paper. If they are the common ones, my concern here is alleviated. Additionally, though, it's challenging to evaluate CSP's performance without comparison to other existing methods. For example, "Recurrent Model-Free RL Can Be a Strong Baseline for Many POMDPs" reports strong performance of RL^2, and VariBAD also reports stronger-than-PEARL results. Inclusion of these methods would help understand whether CSP indeed offers superior performance or not.

**Summary Of The Paper:**

This work proposes to parametrize meta-RL policies with a symbolic policy in the hopes of improving generalization, interpretability, and efficiency.

**Summary Of The Review:**

Overall, this work does not meet or test the claims that it sets out and could significantly be improved in quality and presentation. Therefore, I do not think it currently is ready for publication.

---

### Official Review · Reviewer_SCBW · 2022-10-27

**Confidence:** 2
**Correctness:** 3
**Technical Novelty And Significance:** 3
**Empirical Novelty And Significance:** 3
**Recommendation:** 5

**Clarity, Quality, Novelty And Reproducibility:**

The paper is clearly written and of good quality. Experimental setup is well described and model hyper-parameters are provided, which aids in reproducibility.

**Strength And Weaknesses:**

Strengths

The learned symbolic policies are more interpretable than pure deep learning policies.
The learned symbolic polices have been demonstrated to be lightweight and easier to deploy.

Weaknesses

While the efficiency of inference is shown to be better, no training efficiency is demonstrated. The proposed approach contains two more neural networks to be trained.
The evaluation is only demonstrated on toy examples.

**Summary Of The Paper:**

The paper proposes Contextual Symbolic Policy learning for Meta-Reinforcement Learning, in which a symbolic network represents some search space in the policy space, conditioned on the context represented by encoding previous transitions using an encoder network. In addition, a path generator neural network is trained to select the "correct" symbolic form of the policy, by learning to mask out irrelevant nodes and edges in the symbolic policy network, with the encoded context as input. A parameter generator neural network is also trained, with the encoded context as input, to generate parameters of the symbolic policy network. In general, the whole pipeline, starts with design of the symbolic network architecture (plain, densely connected or arranged densely connected structures), learns to select the appropriate form of the symbolic policy by masking out irrelevant nodes and edges, and also learns the weights on the relevant edges. The pipeline is trained end-to-end.

The proposed approach is evaluated on a select benchmarks Meta-RL tasks and was shown to be better and sometimes competitive to the baselines.

**Summary Of The Review:**

While the paper proposes an approach that allows extraction of symbolic policy expressions and has efficient inference speed, no analysis of the training efficiency is provided, especially given it has additional neural networks that need to be trained. Furthermore, it's not clear if policies of more complex environments will be any interpretable and if performance will scale to large benchmark problems.

The paper claims that by combining different kinds of operators to form the symbolic policy, CSP can handle more complex environments but this is not supported anywhere in the paper.

---

### Official Review · Reviewer_8Fo8 · 2022-10-31

**Confidence:** 4
**Correctness:** 2
**Technical Novelty And Significance:** 2
**Empirical Novelty And Significance:** 2
**Recommendation:** 5

**Clarity, Quality, Novelty And Reproducibility:**

- There are some questions that could be clarified as I mentioned in the main Weaknesses.
- The proposed method is somewhat novel, although it is built on the previous work of (Rakelly et al., 2019) and (Larma et al., 2021).

**Strength And Weaknesses:**

Strengths:
- Symbolic representation is attractive since it has shown some promising in mitigating the issues regarding generalization ability, sample efficiency, and interpretability. This work targets to applying the symbolic representation to the meta RL setting in order to improve the generalization ability, reduce the computational cost, and increase the interpretability. Specifically, a gradient-based learning method is proposed to learn the contextual symbolic policy from scratch in an end-to-end differentiable way, which consists of a symbolic network, a path selector and a parameter generator.


Weaknesses:
- The literature review could include some previous work of combining symbolic representation with reinforcement learning in different ways, such as [1,2,3], although this work focuses on achieving the symbolic representation in a differentiable manner.
- The abuse notation about the context of $c_{T}$: such as $c$ and $c_{\kappa}$.
- In Figure 1: it's not clear where do actor loss and critic loss come from. Is the "context symbolic policy" network only for the policy network? Does $Q(s,a,z)$ represent the value network? If this is the case, what's the architecture for the value network?
- Are the symbolic operators listed in the paper applicable to all different tasks? If not, there should indicate what kinds of tasks can work with this symbolic operator library.
- What's the experimental setup? Are those tasks trained alternatively or separately?
- What are the differences for the paths in Table 2? Currently, I don't understand what are those.
- The empirical evaluation seems weak and the current form of results is somewhat far from what was said to close the gaps in the introduction section, such as overfitting \& poor generalization issue, inefficient deployment issues with limited computational resources, and explainability issue. For example, the problem size looks small. Even for the PEARL, the FLOPs are only in the level of KBytes and this might not be a big issue w.r.t. the deployment. For interpretability, it works well for the classical control problems since the original policy itself has a direct relationship between the state features and the action features. Although the proposed method can learn a symbolic policy, it looks to me that this is only an explicit expression for the learned policy which has partial interpretability rather than only numerical numbers from neural networks. The learned expression could vary if there are more different symbolic operators included.


References:
- [1] Garnelo, M., Arulkumaran, K., & Shanahan, M. (2016). Towards deep symbolic reinforcement learning. arXiv preprint arXiv:1609.05518.
- [2] Garcez, A. D. A., Dutra, A. R. R., & Alonso, E. (2018). Towards symbolic reinforcement learning with common sense. arXiv preprint arXiv:1804.08597.
- [3] Lyu, D., Yang, F., Liu, B., & Gustafson, S. (2019). SDRL: interpretable and data-efficient deep reinforcement learning leveraging symbolic planning. In Proceedings of the AAAI Conference on Artificial Intelligence (Vol. 33, No. 01, pp. 2970-2977).

**Summary Of The Paper:**

This paper focuses on the issues in the neural network-based policy in meta reinforcement learning (RL), such as overfitting \& poor generalization ability, difficult/inefficient to deploy with limited computational resources, and poor interpretability. To address those issues, the framework of Contextual Symbolic Policy (CSP) is proposed by learning a contextual policy with a symbolic form based on the context variables for unseen tasks in meta-RL. Finally, experiments were conducted on several continuous control problems, with results demonstrating its effectiveness in terms of return, FLOPs, and interpretability.


**Summary Of The Review:**

According to my comments in both the main Weaknesses and the section of Clarity, Quality, Novelty, I feel this is a paper where reasons to reject outweigh reasons to accept.

---

### Decision · Program_Chairs · 2023-01-20

**Decision:**

Reject

**Justification For Why Not Higher Score:**

Unanimous decision among reviewers.

**Justification For Why Not Lower Score:**

N/A

**Metareview: Summary, Strengths And Weaknesses:**

I thank the authors for their submission and active engagement during the discussion period. The reviewers unanimously agree that this paper is below the acceptance threshold. In particular, they remark unclear motivation, some missing ablations, issues regarding clarity and novelty. Therefore, I recommend rejection.